# Psychosocial Effects and Use of Communication Technologies during Home Confinement in the First Wave of the COVID-19 Pandemic in Italy and The Netherlands

**DOI:** 10.3390/ijerph18052619

**Published:** 2021-03-05

**Authors:** Sofia Bastoni, Christian Wrede, Achraf Ammar, Annemarie Braakman-Jansen, Robbert Sanderman, Andrea Gaggioli, Khaled Trabelsi, Liwa Masmoudi, Omar Boukhris, Jordan M. Glenn, Bassem Bouaziz, Hamdi Chtourou, Lisette van Gemert-Pijnen

**Affiliations:** 1Centre for eHealth and Wellbeing Research, Department of Psychology, Health and Technology, University of Twente, 7522 NB Enschede, The Netherlands; l.m.a.braakman-jansen@utwente.nl (A.B.-J.); r.sanderman@umcg.nl (R.S.); j.vangemert-pijnen@utwente.nl (L.v.G.-P.); 2Institute of Sport Science, Otto-von-Guericke University, 39106 Magdeburg, Germany; ammar1.achraf@ovgu.de; 3Interdisciplinary Laboratory in Neurosciences, Physiology and Psychology: Physical Activity, Health and Learning (LINP2), UFR STAPS, UPL, Paris Nanterre University, 92000 Nanterre, France; 4Department of Health Psychology, University Medical Center Groningen, 9712 Groningen, The Netherlands; 5Department of Psychology, Universitá Cattolica del Sacro Cuore, 20123 Milan, Italy; andrea.gaggioli@unicatt.it; 6Applied Technology for Neuro-Psychology Lab, I.r.c.c.s. Istituto Auxologico Italiano, 20149 Milan, Italy; 7High Institute of Sport and Physical Education of Sfax, University of Sfax, 3000 Sfax, Tunisia; trabelsikhaled@gmail.com (K.T.); liwa.masmoudi@yahoo.fr (L.M.); omarboukhris24@yahoo.com (O.B.); h_chtourou@yahoo.fr (H.C.); 8Research Laboratory: Education, Motricité, Sport et Santé, EM2S, LR19JS01, High Institute of Sport and Physical Education of Sfax, University of Sfax, 3000 Sfax, Tunisia; 9Exercise Science Research Center, Department of Health, Human Performance and Recreation, University of Arkansas, Fayetteville, NC AR72701, USA; jordan.mckenzie.glenn@gmail.com; 10Multimedia Information Systems and Advanced Computing Laboratory (MIRACL), University of Sfax, 3021 Sfax, Tunisia; bassem.bouaziz@isims.usf.tn; 11Activité Physique, Sport et Santé, UR18JS01, Observatoire National du Sport, 1003 Tunis, Tunisia

**Keywords:** COVID-19, SARS-CoV-2, home confinement, public health, mental wellbeing, loneliness, communication technology

## Abstract

(1) Background: The COVID-19 pandemic forced people from all around the globe to strongly modify their daily routines, putting a significant strain on the social aspects of daily lives. While the first wave of the pandemic was a very challenging time in all countries, it is still uncertain whether various lockdown intensities and infection rates differed regarding their psychosocial impact. This work therefore aimed to investigate (i) the psychosocial effects of home confinement in two European countries that underwent different lockdown intensities: Italy and the Netherlands and (ii) the role of communication technology in relation to feelings of loneliness. (2) Methods: A cross-sectional online survey inquiring about different psychosocial variables and the use of and satisfaction towards communication technology was circulated among the general public during the first wave of the COVID-19 pandemic. In total, 629 participants (66% female, 68% from the Netherlands) answered each question twice, referring to “before” and “during” the pandemic. (3) Results: We found significant negative effects of COVID-19 home confinement on depressive feelings (*p* < 0.001, %∆ = +54%), loneliness (*p* < 0.001, %∆ = +37.3%), life satisfaction (*p* < 0.001, %∆ = −19.8%) and mental wellbeing (*p* < 0.001, %∆ = −10.6%) which were accompanied with a significantly increased need for psychosocial support (*p* < 0.001, %∆ = +17.3%). However, the magnitude of psychosocial impact did not significantly differ between residents undergoing a more intense (Italy) versus a less intense (Netherlands) lockdown, although the decrease in social participation was found to be significantly different for both countries (z = −7.714, *p* < 0.001). Furthermore, our findings demonstrate that the increase in loneliness was associated with the adoption of new digital communication tools (r = 0.21, *p* < 0.001), and significantly higher for individuals who started to adopt at least one new digital communication tool during confinement than for those who did not (z = −4.252, *p* < 0.001). (4) Conclusions: This study highlights that, although COVID-19 home confinement significantly impacted psychosocial wellbeing during the first wave of the pandemic, this impact did not differ based on lockdown intensity. Recognizing the increasing adoption of digital communication technology in an attempt to reduce lockdown loneliness, future studies should investigate what is needed from the technology to achieve this effect.

## 1. Introduction

Coronavirus Disease 2019 (COVID-19) is a viral infection caused by the severe acute respiratory syndrome coronavirus 2 (SARS-CoV-2) [1]. It was first identified in Wuhan, China, in December 2019. In March 2020, based on the spread of the infection, the WHO declared COVID-19 to be characterized as a pandemic [2]. As of 29 January 2021, there have been 100,819,363 confirmed cases of COVID-19 and 2,176,159 deaths, according to WHO [3]. In order to control the rapid spread of the disease and to avoid overloading health systems worldwide, containment measures such as home confinement, limitation of non-essential activities, working remotely and national lockdowns were imposed.

It was initially hypothesized that such strain on people’s lives and disruption of habits and routines might lead to dangerous psychosocial effects. According to recent reviews by Henssler and colleagues [4] and Brooks and colleagues [5], home confinements due to earlier infection outbreaks (SARS, MDR, MRSA, Equine influenza, H1N1, Ebola, MERS) were associated with detrimental psychological effects such as confusion, depression, anxiety, anger, and stress-related disorders. However, in contrast to relatively recent previous infection outbreaks, the current COVID-19 pandemic greatly exceeds earlier quarantine measures on a global scale [6], calling for a need to investigate the psychosocial impact of the COVID-19 pandemic in detail [7]. Recent studies have investigated the impact of COVID-19 home confinement on social, psychological and lifestyle-related outcomes across a mixed sample with participants from Asia, North Africa and Europe [8,9,10,11,12] or within the same country [13]. The work from Moccia and colleagues [13] in particular, gathered early data during the first weeks of the pandemic. The authors found that, within in the first month of lockdown, the Italian general population was already suffering from depressive symptoms. However, to date, less attention has been devoted to how the impact of COVID-19 home confinement compares between different countries and, in particular, between varying lock-down intensities.

Beginning from late March 2020, European countries have introduced national lock-down measures to combat the further spread of COVID-19. In late March, the Netherlands was undergoing the strictest phase of its national containment measures in the first wave of the pandemic. Nonetheless, while Dutch inhabitants were still recommended to stay home as soon as possible, they were still allowed to have limited house guests, to go outside, e.g., for physical activities (not in groups), and have weddings and funeral ceremonies with limited guests [14]. During the same week these measures were introduced, 34 new deaths (22 March 2020) and 537 new cases per day (24 March 2020) were reported in the Netherlands [15]. During the same week, national confinement measures were introduced in Italy as well that largely exceeded those introduced in the Netherlands. As of 22 March, people were not allowed to travel, even by car, to cities in which they did not live. Additionally, travels inside the same city were forbidden, unless motivated by work or health necessities. Non-essential manufactory activities and shops were closed down, outside physical activity of any sort was prohibited, and weddings or funeral ceremonies were disallowed [16]. During the same week Italy was facing 795 new deaths (22 March) and 5.560 new cases per day (23 March) [15]. In conclusion, Italy and the Netherlands adopted two distinct forms of national lock-down strategies, due to the different severity of infection rates and saturation of the health care system during the first half of 2020. Our study took this as a starting point to investigate the psychosocial effects on a population undergoing a “soft” (Netherlands) versus “hard” (Italy) home confinement.

With social distancing measures causing limited social interactions, the use of digital technology plays an increasingly important role by providing virtual opportunities for social connection [17]. Maintaining social networks and safe contact with family and loved ones via digital means has not just been declared as key priority [2], but an inability to do so is associated with longer-term distress [5,18]. Shah and colleagues [17] discussed the construct of lockdown loneliness, and described the feeling of social isolation caused by the enforcement of lockdown measures aimed at containing the spread of the virus. The authors [17] argued that digital communication technology may play a positive role in mediating feelings of lockdown loneliness. Nevertheless, even though numerous digital tools are available, the use and role of digital communication technology in relation to feelings of lockdown loneliness has not yet been sufficiently investigated.

Accordingly, this study aimed to investigate the following questions:What were the psychosocial effects of home confinement during the first wave of the COVID-19 pandemic and how did they differ between countries representing a “hard” (Italy) and “soft” (Netherlands) national lockdown scenario?How did the use of and satisfaction towards digital communication tools change from “before” to “during” home confinement and how did these changes relate to changes in loneliness from “before” to “during” home confinement?

## 2. Materials and Methods

### 2.1. Recruitment and Inclusion Criteria

The data hereby presented was gathered through a cross-sectional online survey starting 19 April 2020, and was collected until 28 June 2020, when containment measures had already eased. Participants were included if they lived in Italy or The Netherlands during the first wave of the pandemic. Subjects under 18 years of age were excluded. Specifically, participants were recruited through snowball sampling: the survey was circulated among institutional and personal networks, via different touchpoints: LinkedIn, Email, Twitter and Facebook. 

### 2.2. Data Privacy and Consent to Participation

The consensus form was included in the survey’s introduction, informing the participants that their data would only be used for research purposes. Furthermore, participants did not have to provide names or contact details. Data was collected according to Google’s privacy policy [19]. The participants were free to interrupt and stop participation in the survey at any time. Partially filled in questionnaires have not been excluded from the analyses. The study received full approval by the Ethics Committees of the University of Twente, Netherlands, the Catholic University of Milan, Italy, and the Otto-von-Guericke University, Germany.

### 2.3. Survey Questionnaires

The online survey used was developed by a multidisciplinary group of experts within our ECLB (Effects of home Confinement onmultiple Lifestyle Behaviors) COVID19 consortium from Germany (Otto-von-Guericke University, University of Münster), Tunisia (University of Sfax) and France (University of Paris-Nanterre). Later, the survey was reviewed and translated by over 50 colleagues including the authors of the present contribution, who have provided the Dutch and Italian versions of the survey and promoted its distribution in the two countries. The ECLB-COVID19 survey included a collection of validated and/or crisis-oriented brief questionnaires. Reliability of adapted questionnaires was tested by the project steering group through piloting, prior to survey administration. These brief crisis-oriented questionnaires showed good to excellent test–retest reliability coefficients (r = 0.84–0.96). The current study reports findings on social participation, mental wellbeing, depressive mood and feelings, life satisfaction, need for psychosocial support, loneliness and the use of and satisfaction towards digital communication tools. In selecting the questionnaires, we relied on three main criteria. Firstly, measures were chosen because they provided coverage of the abovementioned investigated constructs. Secondly, reliability and consistency of the proposed measures was considered. Last, in order to avoid response burden, shorter measures were preferred. Previous studies within our consortium investigated other parts of the survey such as physical activity, diet behavior and general lifestyle and/or reported data from an international sample without a specific focus on comparisons between countries. These findings are published elsewhere [8,9,10,11,12]. In general, all survey questions requested two answers: One referring to the period before and one referring to the period during home confinement. In that way, participants were instructed to compare both periods. The full English version of the questionnaire and the Dutch and Italian versions are available in the Appendix A.

#### 2.3.1. Social Participation

Social participation was measured using the Short Social Participation Questionnaire for Lockdowns (SSPQL) [11]. The tool is a 14-item crisis-oriented adaptation of the Social Participation Questionnaire [20] inquiring the occurrence frequency of several social activities for “before” and “during” home confinement. This questionnaire was translated for Italian and Dutch respondents, following the procedure of translation and backtranslation (Chronbach’s Alpha “before”: 0.637; “during”: 0.498). The total sum-score for the SSPQL ranges from 14 (“never been socially active”) to 70 (“all times socially active”).

#### 2.3.2. Mental Wellbeing

The Short Warwick-Edinburgh Mental Well-being Scale (SWEMWBS) was used to assess mental wellbeing. This 7-item version has recently been validated for the general population [21] and was available in Dutch [22,23] and Italian [24]. Statements are scored on a 5-points scale from “none of the time” (1) to “all of the time” (5). Total scores range from 7 to 35 (7–19.3 = low; 20–27 = medium; 28.1–35 = high) [11] (Chronbach’s Alpha “before”: 0.825; “during”: 0.821).

#### 2.3.3. Depressive Mood and Feelings

Depressive moods and feelings were measured using the short version of the Mood and Feelings Questionnaire (SMFQ) [25], a brief screening tool for depression using a 3-points scale. Scores on the SMFQ range from 0 to 26 with a total score of ≥12 indicating the presence of depressive symptoms [11]. The questionnaire was translated for Italian and Dutch respondents, following the procedure of translation and backtranslation. (Chronbach’s Alpha “before”: 0.900; “during”: 0.895).

#### 2.3.4. Life Satisfaction

In order to assess life satisfaction, the Short Life Satisfaction Questionnaire for Lockdowns vas used (SLSQL) [9] which is a short version of the Satisfaction with Life Scale [26]. Participants answer using a 7-points scale. Total scores for the SLSQL range from 3, indicating the participant is “extremely dissatisfied” to 21, indicating the participant is “extremely satisfied” [9]. The questionnaire was translated for Italian and Dutch respondents, following the procedure of translation and backtranslation (Chronbach’s Alpha “before”: 0.888; “during”: 0.862).

#### 2.3.5. Loneliness and Need for Psychosocial Support

Single-item measures were used to inquire perceived loneliness (“To what extent do you feel lonely?”) and need for psychosocial support (“Do you feel you are in need of psychosocial support?”). Responses were given on a 5-points scale from “none to very small extent” to “ very large extent” (loneliness) and from “never” to “all times” (need for psychosocial support).

#### 2.3.6. Use of and Satisfaction towards Digital Communication Tools

We asked participants to select options they use from a list of several communication tools such as telephone, video calling, social media and messenger apps, and indicate their satisfaction about these on a scale from 1 (“totally unsatisfied”) to 5 (“totally satisfied”). Participants had the possibility to add tools not previously listed.

### 2.4. Data Analysis

The data were analyzed using IBM SPSS statistical software (version 25, IBM Corporation, Armonk, NY, USA). Descriptive statistics were computed and reported as means, standard deviations and change scores (∆ during-before) for all psychosocial- and technology-related outcome variables. Shapiro–Wilk tests were conducted on each outcome variable to test for normality. Since data were not normally distributed, nonparametric tests were applied for statistical analyses. To test for significant within-subjects changes in psychosocial outcomes from “before” to “during” home confinement, Wilcoxon tests were performed. To test whether the magnitude of change in psychosocial outcomes (∆ during-before) significantly varied between countries, Mann–Whitney U tests were used. Effect sizes (Pearson r = z/√N) were calculated to determine the strength of within- and between-subject effects and interpreted using the following criteria: 0.1 ≤ r < 0.3: small; 0.3 ≤ r < 0.5: medium and r ≥ 0.5: large [27]. To test for significant changes in satisfaction towards communication technologies from “before” to “during” home confinement, Wilcoxon signed-rank tests were performed. Furthermore, Spearman correlations were conducted to analyze the relationship between the adoption of new communication technologies and changes in loneliness from “before” to “during” home confinement. Finally, a Mann–Whitney U test was performed to test for significant differences in the change of loneliness from “before” to “during” between people who adopted new technologies (from now on defined as adopters) and people who did not adopt new technologies (from here on: non-adopters). In general, statistical significance was set at α < 0.05.

## 3. Results

### 3.1. Sample description

Table 1 describes the socio-demographic details of participants from Italy and the Netherlands. It is possible to notice that, on average, participants were highly educated (43%) and healthy (85.7%). Additionally, the Italian sample was significantly younger (M = 31.23, SD = 11.11) than the Dutch sample (M = 38.55, SD = 16.25; z = −4.307, *p* ≤ 0.001). Finally, the sample was predominantly female (66.9%). However, Chi square tests showed that countries did not differ significantly regarding gender (χ^2^(2) = 4.667, *p* = 0.097).

### 3.2. Psychosocial Effects of Home Confinement within and between Dutch and Italian Residents

Changes in psychosocial outcome variables from “before” to “during” home confinement in the total sample and per country are presented in Table 2. Results show a significant change within subjects from “before” to “during” home confinement in all tested outcome variables. Scores on depressive mood and feelings increased most by 54% as compared to “before” home confinement, followed by an increase in loneliness (+37.3%) and the need for psychosocial support (+17.3%). Conversely, a decrease was shown in scores on life satisfaction (−19.8%) and mental well-being (−10.6%). Within-subjects effect sizes were large, except for scores on the need for psychosocial support showing a medium effect size.

Changes in social participation scores, in particular, served as a measure to determine whether the Italian and Dutch lockdown measures indeed resulted into different levels of change in social participation. As expected, Italian residents reported a significantly higher decrease in social participation (−44.2%) than Dutch residents (−37.7) (z = −7.714, *p* < 0.001). Otherwise, results revealed no significant difference in psychosocial change from “before” to “during” home confinement between individuals from the Netherlands and Italy.

### 3.3. Changes in Use of and Satisfaction towards Digital Communication Tools in Relation to Loneliness among the General Public

Changes in use of and satisfaction towards digital communication tools from “before” to “during” home confinement among Italy and the Netherlands are presented in Table 3. Descriptive statistics show an increase in usage of video-calling by 31.8%, whereas other communication tools showed minimal increases in usage (ranging between 2% and 0.3%). Furthermore, results show a significant increase in satisfaction towards video-calling (z = −6.508, *p* < 0.001) and a significant decrease in satisfaction towards messenger apps (z = −4.767, *p* < 0.001) from “before” to “during” home confinement.

A small proportion of participants added ways of communication used during confinement which were not mentioned in the questionnaire. Several answers indicated a specification of tools (Whatsapp, Facetime, Zoom, MS Teams, Skype, Instagram), whereas others reported alternatives to technology such as exchanging letters or talking with neighbors “from garden to garden”.

Our analyses involving the variable of loneliness revealed the following results: Spearman correlations showed that the lonelier people felt from “before” to “during” home confinement (M∆ = 0.36, SD∆ = 1.62) the more they started to adopt new digital communication tools (M∆ = 0.63; SD∆ = 1.06) they did not use before confinement already (rs = 0.21, *p* < 0.001). Furthermore, a Mann–Whitney U test showed that the increase in loneliness was significantly higher for participants whom started to adopt at least one new digital communication tool during home confinement than for those who did not (z = −4.252, *p* < 0.001).

## 4. Discussion

While the detrimental effects of natural disasters [28,29] and the consequences of isolation measures due to illness and quarantine [4] have been widely studied, what is known in literature is only partly applicable to the COVID-19 pandemic due to its undetermined duration and diffusion worldwide [6]. The global populace has been finding themselves in a situation of constant effort to adjust to home confinement and a new way of living that is completely uncertain and constantly changing. As a result, more evidence was needed in order to fully capture the magnitude of psychosocial consequences of home confinement during this pandemic. Furthermore, although COVID-19 is a worldwide problem, different countries have faced very different extents of containment measures and severity of infection rates. Therefore, this investigation is focused specifically on psychosocial impact in Italy, one on the European countries that was most hardly hit by the pandemic during the first wave, and the Netherlands that faced a less intense lockdown. In times of social distancing circumstances, we furthermore aimed to investigate changes in use of and satisfaction towards digital communication tools and how these related to possible changes in loneliness.

The main findings of the present work include that, as hypothesized, there has been a significant worsening in all psychosocial outcome variables under investigation. This result is in line with both general literature around epidemic situations [4] and recent studies about the COVID-19 pandemic [30]. Particularly, higher loneliness, need for psychosocial support and depressive mood and feelings were found both in Italy and the Netherlands. Conversely, our participants reported significantly lower scores on mental well-being and life satisfaction. However, while the Netherlands suffered a less intense lockdown and lower infection rates [14,16], there were no significant differences between the two countries on any of the studied psychosocial outcome variables. The only significant difference between countries was shown in social participation, revealing that the lockdown in Italy resulted in a significant stronger reduction of social contacts (self-reported) compared to the Netherlands. These findings suggest that the severity of lockdown measures and infection rates do not have effect on the psychosocial impact of a pandemic. The present study focused on life satisfaction, depressive mood and feelings, loneliness, need for psychosocial support and mental wellbeing. While our study highlights the urgency to monitor and mitigate these outcomes, other related social and/or psychological consequences and detrimental effects of the pandemic should not be overlooked. For instance, recent studies have emphasized the occurrence of disease stigma and discrimination among COVID-19 patients [31] and bullying and harassment against healthcare workers [32]. Furthermore, Rooksby and colleagues [33] argue that, especially amongst young adults, there is higher risk of developing a Hikikomori syndrome. Specifically, this mental health issue seems to be related to the failure or impossibility to achieve otherwise expected goals and milestones. This (im) possibility might have been increased by the isolation and limitations related to the outbreak. These studies [31,32,33] represent a number of examples within a wide range of possible psycho(social) consequences of the COVID-19 pandemic which have not yet been completely identified or studied. Other than studying different outcomes, research has also focused on how aware people are about the danger they are encountering. For example, risk perception is particularly relevant in the public health domain, because it is related with the adoption of preventative health behaviors [34] and carries implications for effective (governmental) risk communication [34].

One of the many changes from the pandemic involved people increasing the number of tasks they performed “online”. In fact, work, education, and even caregiving duties were forced in a new connected way of taking place. Most importantly, reliance on digital communication tools was the first and fastest response to continue working from home, but also for maintaining social contacts. However, the protective role of communication technologies is still up for debate. If on one hand, authors believe that technology could mitigate the feeling of loneliness, and more specifically lockdown loneliness [17], the literature is rich with studies about the detrimental effects that technology-mediated communication could have had on interpersonal relationships during COVID-19 [35]. The present contribution provides needed knowledge about changes in the use of and satisfaction towards digital communication tools and their relations with changes in loneliness. Specifically, inquiries were made about the use of telephone, video-calling, messenger apps and social media. Our results show that technology usage increased during the pandemic, indicating a greater reliance on technology to maintain safe contact. Satisfaction significantly increased for video calling, while it decreased for messenger apps, suggesting a preference for more immersive tools [36]. In this sense, contact through video calling might be seen as more satisfying than through messenger apps. In fact, Hietanen and colleagues [37] suggest that eye contact through video calling could help overcoming the lack of physical presence. Furthermore, our findings showed the increase in loneliness was associated with the adoption of new digital communication tools. In fact, the increase was significantly higher for individuals who started to adopt new digital tools during home confinement than for those who did not. This suggests participants used communication technology in an attempt to reduce loneliness. However, insights into the effect of using communication technology on loneliness cannot be provided by our cross-sectional study. Future research should use rigorous methods to investigate in how far and under which circumstances such effects can occur, especially among vulnerable groups such as elderly [38].

### Strengths and Limitations

The primary strength of this large-scale study is that the data was collected very quickly during the COVID-19 pandemic restrictions using a fully anonymous survey including validated crisis-oriented questionnaires. However, even though the present survey covered a wide range of psychosocial outcomes, data on other variables such as work-related stress, burnout, stigma and discrimination were not collected. Other principal limitations of the present study include the different sample sizes in Italy and the Netherlands, as well as differences in terms of age and education level. Although the difference is not significant among countries, the sample is predominantly female. Furthermore, respondents that classified their health status as “other” (only 1.6% of the whole sample) might suffer from psychiatric diseases. These can act as confounding variables. Furthermore, limitations of the cross-sectional self-report mode should be considered: Given the unexpected worldwide outbreak of the COVID-19 pandemic, responses on outcome variables referring to T0 (before the pandemic) were based on subjective recall of participants. Even more so, they were asked to reflect on their prior living situation when the pandemic was still in place, leading to the risk of an overestimation of the positivity of their prior condition. Lastly, the present study was not designed with the purpose of recontacting the participants periodically or after vaccination. Future research should use designs that allow for longitudinal analysis.

## 5. Conclusions

The present study concludes that, although COVID-19 home confinement significantly impacted psychosocial wellbeing, this impact did not differ based on lockdown intensity. This result implies that, apart from the health emergency and the economic crisis that will follow, the psychosocial emergency should also be addressed in all countries, and not only in those who were most severely affected by the pandemic. On a practical level, health policy makers working on prevention or mitigation of the detrimental consequences of home-confinement should focus on trying to raise awareness of the negative consequences in counties that faced more severe lock-downs, but not neglect countries that underwent lighter limitations. Additionally, addressing depressive feelings and loneliness should be prioritized. Our consortium has previously published recommendations to overcome the negative effects of home confinement [39,40] including innovative information and communication technology-based concepts to provide crisis-oriented health surveillance- and recommendations for extra vulnerable groups such as elderly [41]. To create and evaluate possible mitigation strategies for the current or any future pandemic, it will be essential to keep monitoring psychosocial consequences of home confinement longitudinally. Finally, recognizing an increasing adoption of digital communication technology in an attempt to reduce lockdown loneliness, future studies should investigate if and how such an effect can best be achieved, creating useful requirements for development. On a practical level, this knowledge might help lay the groundsworks for technological interventions that safely mitigate feelings of loneliness during the pandemic.

## Figures and Tables

**Table 1 ijerph-18-02619-t001:** Socio-demographic overview Italy and The Netherlands.

Variable	Category	Total *(N = 629)*		The Netherlands *(N = 427)*		Italy *(N = 202)*	
		*M*	*SD*	*M*	*SD*	*M*	*SD*
Age	(Years)	36.20	15.18	38.55	16.25	31.23	11.11
		***N***	***%***	***N***	***%***	***N***	***%***
Gender	Male	206	32.8	128	30.0	78	38.6
	Female	420	66.9	297	69.6	123	60.9
	Other	3	0.5	2	0.5	1	0.5
Marital status	Single	312	49.6	185	43.3	127	62.9
	Married/Living as a Couple	293	46.6	224	52.5	69	34.2
	Widowed/Divorced/Separated	24	3.8	18	4.2	6	3.0
Level of Education	No Schooling complete	6	1	-	-	6	3.0
	High School Diploma/Equivalent	88	14	27	6.3	61	30.2
	Professional Degree	62	9.9	55	12.9	7	3.5
	Bachelor’s Degree	198	31.5	144	33.7	54	26.7
	Master/Doctorate degree	275	43.7	201	47.1	74	36.6
Employment status	Employed for wages	293	46.6	215	50.4	78	38.6
	Self-employed	48	7.6	17	4.0	31	15.3
	Out of work/Unemployed	11	1.7	7	1.6	4	2.0
	Student	148	23.5	96	22.5	52	25.7
	Retired	35	5.6	34	8.0	1	0.5
	Unable to work	8	1.3	7	1.6	1	0.5
	Problem caused by COVID-19	12	1.9	2	0.5	10	5.0
	Other	70	11.1	45	10.5	25	12.4
Health status	Healthy	539	85.7	353	82.7	186	92.1
	Risk Factor for Cardiovascular Disease	62	9.9	50	11.7	12	5.9
	Cardiovascular Disease	10	1.6	9	2.1	1	0.5
	With Cognitive Impairment	5	0.8	4	0.9	1	0.5
	Other	10	1.6	8	1.9	2	1.0

Note. Need for psychosocial support (N = 629); Loneliness (N = 487); remaining variables (N = 581).

**Table 2 ijerph-18-02619-t002:** Psychosocial effects of COVID-19 home confinement within and between Dutch and Italian residents.

Variable	Range	Total			Within-Subjects Effects			NL	ITA	Between-Subjects Effects		
		Before M (SD)	During M (SD)	∆ (%∆)	*z* Value	*p* Value	Effect Size *r*	∆ (%∆)	∆ (%∆)	*z* Value	*p* Value	Effect Size *r*
Social participation	14–70	41.97 (7.48)	25.13 (4.86)	−16.84 (−40.8)	−20.871	<0.001	0.866	−15.20 (−37.7)	−19.91 (−44.2)	−7.714	<0.001	0.320
Mental wellbeing	7–35	27.04 (3.43)	24.17 (4.51)	−2.87 (−10.6)	−15.635	<0.001	0.649	−2.82 (−10.3)	−2.98 (−11.4)	−0.423	0.672	0.018
Depressive mood and feelings	0–26	4.74 (4.76)	7.30 (5.68)	+2.56 (+54)	−13.523	<0.001	0.561	+2.63 (+78.5)	+2.42 (+32.9)	−1.236	0.216	0.051
Life satisfaction	3–21	15.73 (3.25)	12.62 (4.50)	−3.12 (−19.8)	−16.334	<0.001	0.678	−3.10 (−18.8)	−3.15 (−21.9)	−0.128	0.898	0.005
Loneliness	1–5	1.69 (0.81)	2.33 (1.31)	+0.63 (+37.3)	−11.297	<0.001	0.512	+0.67 (+41.9)	+0.57 (+30.2)	−0.811	0.417	0.037
Need for psychosocial support	1–5	1.73 (0.84)	2.03 (1.03)	+0.30 (+17.3)	−10.133	<0.001	0.404	+0.28 (+16.8)	+0.33 (+17.6)	−0.543	0.587	0.022

Note. Need for psychosocial support (N = 629); Loneliness (N = 487); remaining variables (N = 581).

**Table 3 ijerph-18-02619-t003:** Use of and satisfaction towards digital communication tools before and during home confinement among the general public (N = 629).

Tools	Usage		Satisfaction					
	Before	During	Before		During		Change	
	Used by (%)	Used by (%)	M	SD	M	SD	%∆	*p*
Telephone	97.1	97.3	4.32	0.79	4.18	0.84	−3.2	0.125
Video-calling	58.5	90.3	3.88	0.96	4.13	0.90	+6.4	<0.001
Social media	81.1	83	4.05	0.87	3.98	0.95	−1.7	0.339
Messenger apps	78.9	80.9	4.29	0.89	4.20	0.91	−2.1	<0.001
Total satisfaction			4.13	0.68	4.10	0.76	−0.73	
∑ tools used per person (M, SD)	3.16 (1.01)	3.52 (0.80)						

## Data Availability

During the informed consent procedure, participants were assured that data would only be used for research purposes and not be publicly available as advised by the Otto-von-Guericke University Ethics Committee. The data are therefore available from the following authors: S.B. (s.bastoni@utwente.nl), C.W. (c.wrede@utwenten.nl) and A.A. (ammar1.achraf@ovgu.de) upon reasonable request related to research purposes such as validation, replication, re-analysis, new analysis, re-interpretation or inclusion into meta-analyses.

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
