# Peer review of "Psychosocial Effects and Use of Communication Technologies during Home Confinement in the First Wave of the COVID-19 Pandemic in Italy and The Netherlands"

_ijerph, 2021, doi:10.3390/ijerph18052619_

Round 1

Reviewer 1 Report

Dear Authors,

topics covered in your article are of current interest in relation to the widespread of COVID-19 pandemic and sharing your experience can provide a contribution to a better understanding of psychological impact during disease outbreaks. The concepts overall are sufficiently well and logically developed. Methods and results are clearly described and well reported.

I would suggest better characterizing the clinical information of participants and specifying whether the group "Other" also includes psychiatric diseases, since their presence might influence the sample presentation and the results. Alternatively, if such information is not available, authors are encouraged to add it as a limit.

Moreover, the background of the study and the discussion section should be improved and references on the topic widened.

In this view, I would suggest commenting on the following article “Moccia L et al. Affective temperament, attachment style, and the psychological impact of the COVID-19 outbreak: an early report on the Italian general population. Brain Behav Immun. 2020;87:75-79. doi:10.1016/j.bbi.2020.04.048”, to report other data about COVID-19 psychological consequences on the Italian general population. Last, authors might benefit from eventually referring to “Di Nicola et al. Serum 25-hydroxyvitamin D levels and psychological distress symptoms in patients with affective disorders during the COVID-19 pandemic. Psychoneuroendocrinology. doi.org/10.1016/j.psyneuen.2020.104869”, who investigated psychological distress and clinical worsening of affective symptoms in relation to COVID-19 pandemic. More specifically, discussion could benefit from a deeper description of the relationship between mental health problems and COVID-19 outbreak giving space to a wider range of potential risk factors, also biological ones (such as the serum levels of Vitamin D), in addition to the more commonly debated socioeconomic predictors.

Best regards.

Reviewer 2 Report

Dear Authors,

the impact of the current pandemic is transversal to human nature itself, affecting health and well-being at all levels.

Even the world of work, as we have been used to knowing and perceiving it, has suffered a strong impact in the current pandemic scenario for over a year to date.
Even the most atavistic, inert and less prone to change cultures, such as those of the Old Continent, have had to launch work from home. A forced and sudden change that has taken employers and workers off guard, in some States rather than in others.

Thank you for sharing your experience.

Paper needs a finishing touch and some insights in order to raise the bar of your paper.

  • why not SARS-CoV-2 among keywords? why not work from home?
  • please try to contextualize and deepen the current pandemic scenario when introducing your work;
  • which criteria for the selection of psychometric tools?
  • are you sure that you have contemplated all the psychosocial risks that may lurk in home confimento or in working from home? In such a peculiar scenario you should consider other factors: what about work-related stress, burnout, stigma and discrimination, FOMO?! You have to deal with other psi-factors;
  • what about risk perception? Do you think we can hypothesize, or in any case investigate, a hermit syndrome or hikikomori
  • please report link to appendices, I'm afraid I cannot reach them;
  • results seem to be valid, but so little used in the discussion, that must be improved;
  • you have to deal with gender, it's a good insight even if not statistically significant;
  • deepen point by point your result outcomes;
  • discussions can generally be improved;
  • conclusions must be improved. What are the possible repercussions? What suggestions to give to the health policy maker? Define a clear "take home message" from your perspective and address a conclusion section. You need to take a leap forward;
  • please state in the conclusion if you will re-contact participants to retake the questionnaire after the pandemic or after being vaccinated;

It would be interesting to understand if these people, also in correlation to the perception of risk, consider vaccination important or would hesitate (please also refer to https://www.who.int/news-room/spotlight/ten-threats-to-global-health-in-2019 ).

You can significantly improve your paper.

Please update these gaps referring to the following references:

  • Irigoyen-Camacho, M.E.; Velazquez-Alva, M.C.; Zepeda-Zepeda, M.A.; Cabrer-Rosales, M.F.; Lazarevich, I.; Castaño-Seiquer, A. Effect of Income Level and Perception of Susceptibility and Severity of COVID-19 on Stay-at-Home Preventive Behavior in a Group of Older Adults in Mexico City. Int. J. Environ. Res. Public Health 2020, 17, 7418
  • Baldassarre, A.; Giorgi, G.; Alessio, F.; Lulli, L.G.; Arcangeli, G.; Mucci, N. Stigma and Discrimination (SAD) at the Time of the SARS-CoV-2 Pandemic. Int. J. Environ. Res. Public Health 2020, 17, 6341
  • Sarah Dryhurst, Claudia R. Schneider, John Kerr, Alexandra L. J. Freeman, Gabriel Recchia, Anne Marthe van der Bles, David Spiegelhalter & Sander van der Linden (2020) Risk perceptions of COVID-19 around the world, Journal of Risk Research, DOI: 10.1080/13669877.2020.1758193
  • Wong, B.Y.-M.; Lam, T.-H.; Lai, A.Y.-K.; Wang, M.P.; Ho, S.-Y. Perceived Benefits and Harms of the COVID-19 Pandemic on Family Well-Being and Their Sociodemographic Disparities in Hong Kong: A Cross-Sectional Study. International Journal of Environmental Research and Public Health 2021, 18, 1217
  • Weinstein, B.; da Silva, A.R.; Kouzoukas, D.E.; Bose, T.; Kim, G.J.; Correa, P.A.; Pondugula, S.; Lee, Y.; Kim, J.; Carpenter, D.O. Precision Mapping of COVID-19 Vulnerable Locales by Epidemiological and Socioeconomic Risk Factors, Developed Using South Korean Data. International Journal of Environmental Research and Public Health 2021, 18, 604
  • Dye, T.D.; Alcantara, L.; Siddiqi, S.; Barbosu, M.; Sharma, S.; Panko, T.; Pressman, E. Risk of COVID-19-related bullying, harassment and stigma among healthcare workers: an analytical cross-sectional global study. BMJ Open 2020, 10, e046620
